# STR Profiling Reveals Tumor Genome Instability in Primary Mediastinal B-Cell Lymphoma

Natalya Risinskaya [1], Yana Mangasarova [1], Elena Nikulina [1], Yana Kozhevnikova [2], Julia Chabaeva [1], Anna Yushkova [1], Aminat Magomedova [1], Sergey Kulikov [1], Hunan Julhakyan [1], Sergey Kravchenko [1] and Andrey Sudarikov [1,*]

[1] National Medical Research Center for Hematology, Ministry of Health of Russian Federation, 125167 Moscow, Russia; risinska@gmail.com (N.R.); v.k.jana@mail.ru (Y.M.); lenysh2007@rambler.ru (E.N.); uchabaeva@gmail.com (J.C.); ann.unikova@bk.ru (A.Y.); maminat@mail.ru (A.M.); smkulikov@mail.ru (S.K.); oncohematologist@mail.ru (H.J.); skkrav@mail.ru (S.K.)
[2] School of Medicine, Lomonosov Moscow State University, 119991 Moscow, Russia; kozh.yana@mail.ru
* Correspondence: dusha@blood.ru

**Abstract:** Primary mediastinal B-cell lymphoma (PMBCL) is the only non-Hodgkin's lymphoma variant responding to immune checkpoint inhibitor (ICI) therapy, approximately in half of the cases; however, no molecular markers predicting a response to ICI therapy in PMBCL have been described so far. In this study, we assessed the incidence of the loss of heterozygosity (LOH), elevated microsatellite alteration at selected tetranucleotides (EMAST), and microsatellite instability (MSI) in the tumor genomes of 72 patients with PMBCL undergoing high-dose chemotherapy treatment at the National Research Center for Hematology (Moscow, Russia). Tumor DNA was isolated from biopsy samples taken at diagnosis. Control DNA was isolated from the blood of patients in complete remission or from buccal epithelium. STR-profiles for LOH and EMAST were assessed by PCR with COrDIS Plus multiplex kit (Gordiz Ltd., Moscow, Russia). LOH was detected in 37 of 72 patients (51.4%). EMAST was found in 40 patients (55.5%); 24 had a combination of EMAST with LOH. MSI-high was not found, while MSI-low was detected only in one patient. The association of certain genetic lesions with the clinical outcome in patients receiving treatment according to the standard clinical protocol R-Da-EPOCH-21 has been estimated (58 patients out of 72) and no associations with the worst overall or event-free survival were found.

**Keywords:** PMBCL; LOH; EMAST; MSI; microsatellite stability (MSS)

## 1. Introduction

Primary mediastinal B-cell large cell lymphoma (PMBCL) accounts for 2–4% of all non-Hodgkin's lymphomas and is more common in young women [1,2]. The classical clinical feature of PMBCL at the onset of the disease is the involvement of the anterior-superior mediastinum, with a spread to the surrounding organs and tissues. PMBCL as an independent variant of diffuse large B-cell lymphoma (DLBCL) was defined primarily on clinical data, and further studies, in particular, exome sequencing and an assessment of the gene expression profiles, fully confirmed the validity of this definition. PMBCL tumor cells are characterized by a unique gene expression profile, mutational landscape, and putative driver genes, which are different from all variants of DLBCL and bear great similarity to the classic Hodgkin's lymphoma patterns [3,4]. The PMBCL tumor cells' ability to "escape" from the immune surveillance in the thymic microenvironment is associated with genetic lesions protecting the tumor from T-cell recognition. A significant amount of data on the immune surveillance mechanisms that prevent the development of malignant neoplasms and the role of immunological tolerance in tumor progression and dissemination have recently been accumulated. The discovery of immune checkpoints inhibitors (ICIs), a family of receptors and ligands that are essential for the immune response modulation [5–7],

provides additional insight into the development of tumors and suggests new treatment approaches [8,9]. The use of modern immunotherapeutic drugs makes it possible to achieve a stable clinical response in some patients with various solid tumors [10]. PD-1/PD-L1 blockade therapy is also approved for patients with two hematological malignancies. These are classical Hodgkin lymphoma patients who have relapsed or are refractory after ≥3 lines of therapy (69% overall response rate to pembrolizumab reported) and PMBCL patients who have relapsed after ≥2 lines of therapy (overall response rate to pembrolizumab was 45%) [11,12]; however, despite good results, a resistance to ICI therapy was reported for a significant number of cases. In this regard, the search for prognostic and predictive molecular markers to identify a subgroup of patients that may benefit from ICI therapy is of great practical value [13].

The loss of heterozygosity (LOH) in short tandem repeat (STR) loci, elevated microsatellite alteration at selected tetranucleotides (EMAST) and microsatellite instability (MSI) are intrinsic features of diverse tumor cells [14–16]. MSI-high (at least two unstable microsatellites from pentaplex BAT-25, BAT-26, NR-21, NR-24, and NR-27) is associated with a good response to immunotherapy for numerous solid tumors [17,18].

For PMBCL, these molecular markers have not yet been proven to have prognostic value [3,4,13]. It should be noted that the standard set of markers usually applied for MSI analysis in solid tumors, might not be informative in aggressive lymphoma cases since there is a low occurrence (up to 5%). Furthermore, in about half of the cases, polymorphic inherited allele may also be present in the control material, characterizing not the tumor, but the genetic features of the patient [19,20]. According to the literature data, an EMAST-high is associated with an MSI-high in solid tumors [21]. It should be noted that same the DNA reparation enzymes participate both in mismatch and insertion-deletion reparation. Therefore, certain defects in the DNA repair complex (MMR) may lead to both MSI and EMAST. In this regard, LOH and EMAST in STR loci in the tumor genome are of interest, therefore, an appropriate analysis of these features might be used for the selection of patients for ICI therapy. Our aim was to compare the frequencies of LOH, EMAST, and MSI in the tumor cells of PMBCL patients at diagnosis and to check the possible associations with disease features.

Standard chemotherapy protocols allow achieving a high remission rate in patients with PMCL. The rare resistant patients are candidates for high-dose chemotherapy and autologous hematopoietic stem cell transplantation (auto-HSCT). It can be assumed that the use of ICI would allow a number of patients to avoid the second line of intensive high-dose chemotherapy. Establishing the molecular predictors of response to ICI will make it possible to personalize the therapy protocol and make it as effective as possible.

## 2. Materials and Methods

STR profiles of the tumor cell DNA were analyzed on a cohort of 72 de novo diagnosed PMBCL patients (24 male, 48 female, age 18–58, median age 32) undergoing R-Da-EPOCH-21 (58 patients) and R-m-NHL-BFM-90 (14 patients) regimen treatment at the National Research Center for Hematology (Moscow, Russia).

Tumor DNA was isolated from the biopsy samples taken at diagnosis. Control DNA was isolated from the blood samples of patients taken in complete remission or from buccal epithelium. LOH and EMAST patterns were analyzed using a COrDIS Plus multiplex PCR kit for the amplification of 19 STR loci (D3S1358, D5S818, D7S820, D8S1179, D13S317, D16S539, D18S51, D21S11, CSF1PO, FGA, TH01, TPOX, VWA, D1S1656, D2S441, D10S1248, D12S391, D22S1045, and SE33) and amelogenin X and amelogenin Y loci (Gordiz Ltd., Moscow, Russia). MSI targets (BAT-25, BAT-26, NR-21, NR-24, and NR-27) were evaluated by a COrDIS MSI kit (Gordiz Ltd., Moscow, Russia). An ABI-3130 sequencer (Applied Biosystems, Waltham, Massachusetts, USA) was used for the fragment analysis of PCR products.

For the 58 patients undergoing R-Da-EPOCH-21, the prognostic impact of genetic instability on the clinical outcome was evaluated by survival endpoints: overall survival

(OS), and event-free survival (EFS). Time for OS was defined as an interval from the diagnosis to death; censoring was completed at the date of the last contact. The time for EFS was defined as an interval from the diagnosis to the first adverse event (relapse or refractory disease or death) and the censoring was completed on the date of the last contact [22]. The groups were compared by Kaplan–Maier estimators and a log-rank test.

## 3. Results

LOH in the STR loci was detected in 37 of 72 patients (51.4%); LOH 2p14 and LOH 12p13.2 were most frequent. EMAST was found in 40 patients (55.5%), and SE33 (6q) was the most unstable STR marker (Figure 1).

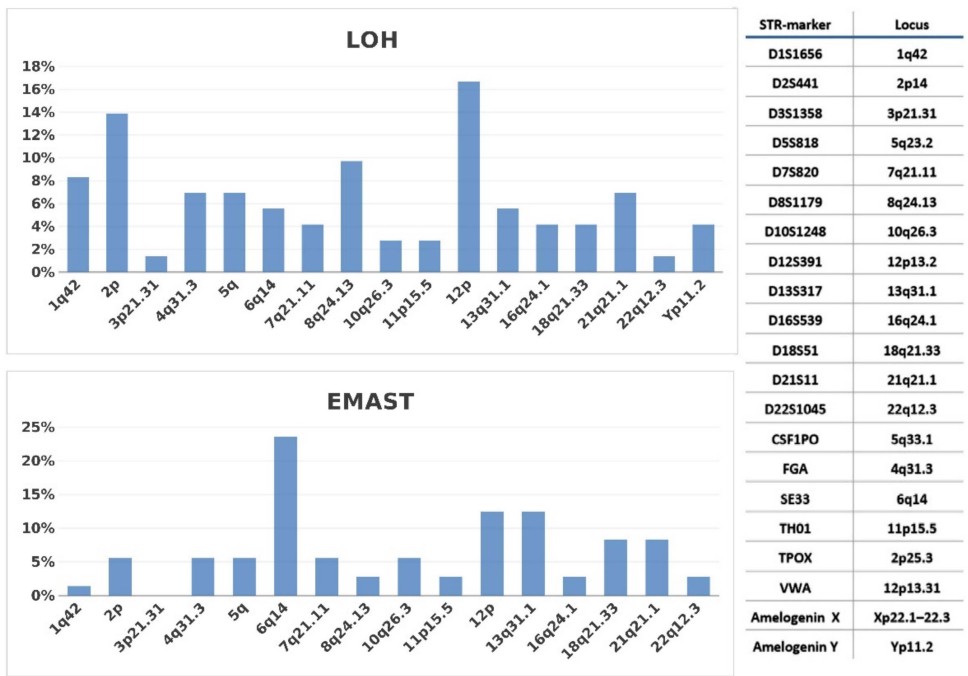

**Figure 1.** Diagram of LOH and EMAST distribution in STR-loci among 72 PMBCL patients.

Twenty-four patients had a combination of EMAST with LOH. Samples of the LOH and EMAST aberrations in the STR loci are presented below (Figure 2).

For all EMAST-positive patients, the number of involved loci was no more than 4 out of 19. The most LOH-positive patient had 10 LOH in STR loci at the 3p, 11p, 12p, 1q, 4q, 13q, 18q, 8q, 6q, and 21q arms of chromosomes. Previously, sequencing of the clinical exome in this patient revealed a c.1896A>G mutation in the MSH3 gene. It should be noted, that MSH3 is the only enzyme in the DNA repair complex that is not involved in the repair of single mismatched nucleotides. Therefore, its deficiency can explain the high occurrence of LOH and EMAST in the absence of MSI. The study of the association of MSH3 gene mutations with the number of aberrant STR loci can provide us with a simple method for assessing MMR deficiency in lymphomas by the instability of the tumor STR profile.

No consensus criteria for the discrimination between groups with different levels of EMAST and LOH has been set so far. In this study, we treated three or more events as a "high" occurrence, based on the 75% percentile or Q3. For less than 25% of patients in this cohort, the presence of LOH was found in three or more markers. Similarly, less than 25% of patients had EMAST in tumor DNA in three or more markers. Figure 3a shows the distribution of patients according to the presence, absence, or combination of factors of microsatellite instability. Figure 3b—distribution of "high" or "low" occurrence of the lesions. The "low" occurrence group included all patients without any sign of genetic instability and patients with one or two unstable markers for each test (LOH, EMAST, MSI).

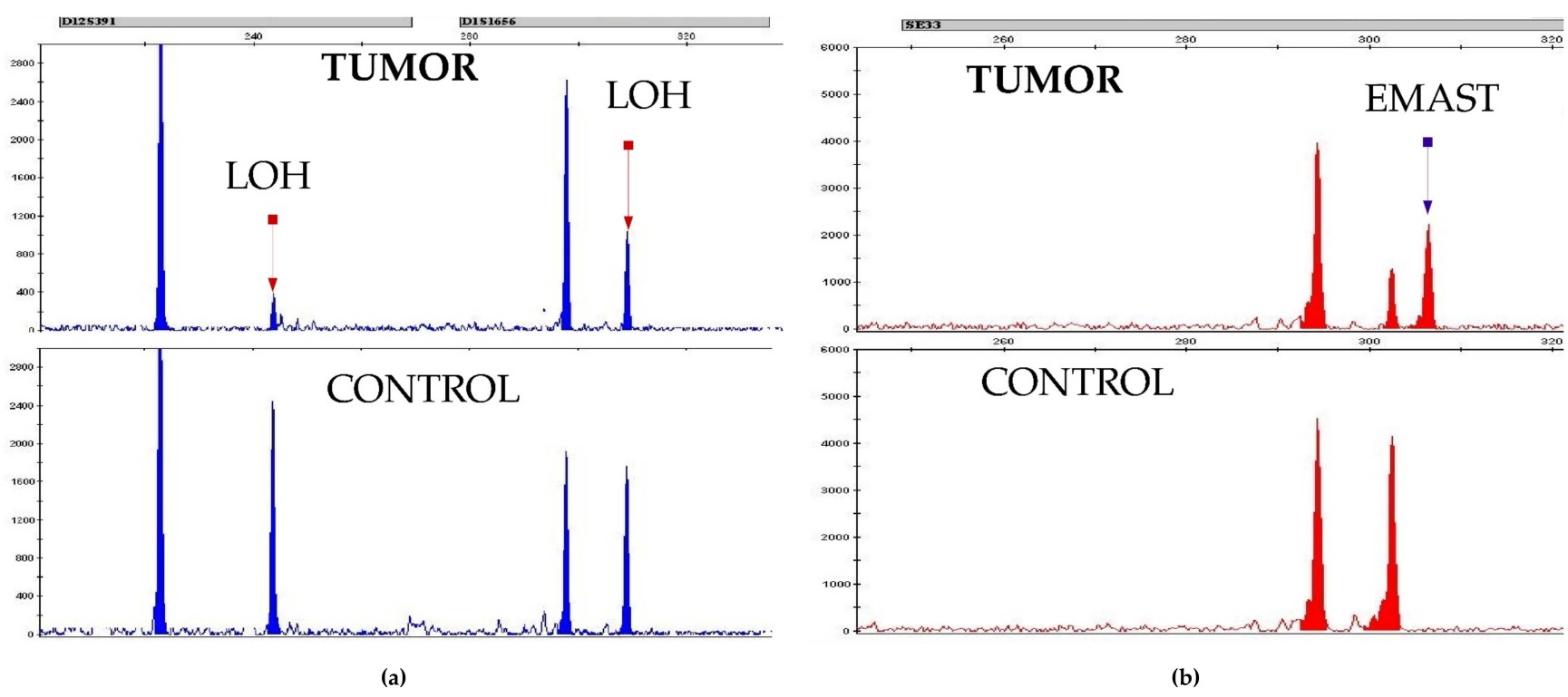

**Figure 2.** LOH and EMAST patterns in STR profiles of PMBCL patients: (**a**) two LOH aberrations in tumor DNA; (**b**) EMAST in tumor DNA.

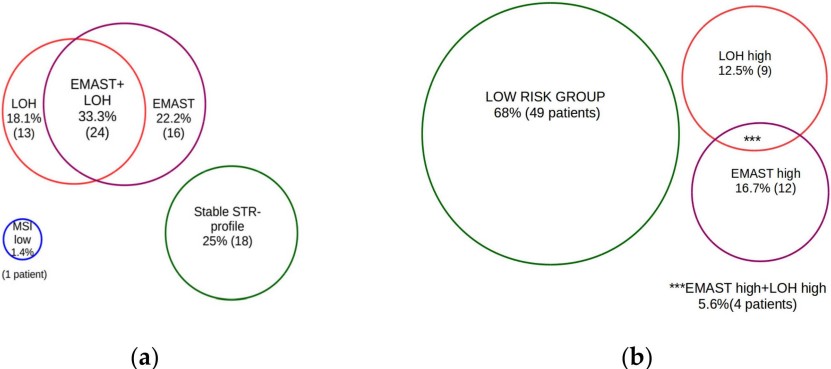

**Figure 3.** Venn diagram indicating the distribution of 72 PMBCL patients according to their LOH, MSI and EMAST status. (**a**) distribution according to the presence, absence, or combination of lesions (**b**) distribution of "high" or "low" occurrence of the lesions.

The majority of patients (64) showed a quasi-homozygous quasi-monomorphic microsatellite profile according to BAT-25, BAT-26, NR-21, NR-24, and NR-27 marker status (Figure 4a). The MSI-high phenomenon was not detected at all. One patient was found to have MSI-low in regard to the NR27 marker (Figure 4b), and his tumor STR profile did not differ from his control DNA STR profile. In five patients, the second allele of the BAT-25 microsatellite marker was detected in both the tumor and control DNA; the same was detected in two patients with the NR-21 marker (Figure 4c). These patients also belonged to the MSS group, since they had a heterozygous marker, which is a rare variant of the norm. Due to the minimal incidence of MSI in PMBCL patients, we excluded this factor from consideration when assessing the overall and event-free survival.

The association of LOH and EMAST status of patients with overall and event-free survival was assessed in a cohort of 58 patients receiving therapy according to the same R-Da-EPOCH-21 protocol, thereby excluding the effect of different therapy options on the outcome of the disease. We have checked the distribution of all major risk factors such as age, gender, LDG level, age-adjusted International Prognostic Index (aa IPI), extramediastinal disease, bulky disease, involvement of the pleura/pericardium, and complete remission/refractory disease with accordance to LOH and EMAST status. No essential differences between the groups were found (Table 1).

**Table 1.** The main characteristics of patients receiving R-Da-EPOCH-21 therapy and their LOH and EMAST status.

| Parameters | PMBCL | LOH-Positive | LOH-Negative | P χ2 | EMAST-Positive | EMAST-Negative | P χ2 |
|---|---|---|---|---|---|---|---|
| *n* | 58 | 27 (47%) | 31 (53%) | | 35 (60%) | 23 (40%) | |
| Male:Female | 19:39 | 10:17 | 9:22 | 0.518 | 11:24 | 8:15 | 0.791 |
| Age, median | 33 (20–58) y | 32 (21–52) y | 33 (20–58) y | | 31 (21–58) y | 33 (20–52) y | |
| [1] LDH<br>N<br>↑ N | <br>4 (7%)<br>54 (93%) | <br>2 (8%)<br>25 (92%) | <br>2 (7%)<br>29 (93%) | <br><br>0.887 | <br>4 (12%)<br>31 (88%) | <br>0<br>23 (100%) | <br><br>0.093 |
| [2] aa IPI<br>0<br>1<br>2<br>3 | <br>1 (2%)<br>2 (3%)<br>47 (81%)<br>8 (14%) | <br>1 (4%)<br>1 (4%)<br>20 (74%)<br>5 (18%) | <br>0<br>1 (3%)<br>27 (87%)<br>3 (10%) | <br><br><br>0.207 | <br>1 (3%)<br>2 (6%)<br>27 (77%)<br>5 (14%) | <br>0<br>0<br>20 (87%)<br>3 (17%) | <br><br><br>0.352 |
| Extramediastinal disease | 9 (15%) | 6 (22%) | 3 (10%) | 0.189 | 6 (17%) | 3 (13%) | 0.694 |
| Bulky disease | 57 (98%) | 27 (100%) | 30 (96%) | 0.347 | 34 (97%) | 23 (100%) | 0.414 |
| Involvement pleura/pericardium | 41 (71%) | 20 (74%) | 21 (67%) | 0.598 | 24 (68%) | 17 (74%) | 0.662 |
| [3] CR<br>refractory disease | 52 (90%)<br>6 (10%) | 23 (85%)<br>4 (15%) | 29 (93%)<br>2 (7%) | <br>0.297 | 30 (86%)<br>5 (14%) | 22 (97%)<br>1 (3%) | <br>0.225 |

[1] LDH—lactatdehydrogenase (N—normal levels, ↑ N—elevated levels), [2] aa IPI—age-adjusted International Prognostic Index, [3] CR—complete remission.

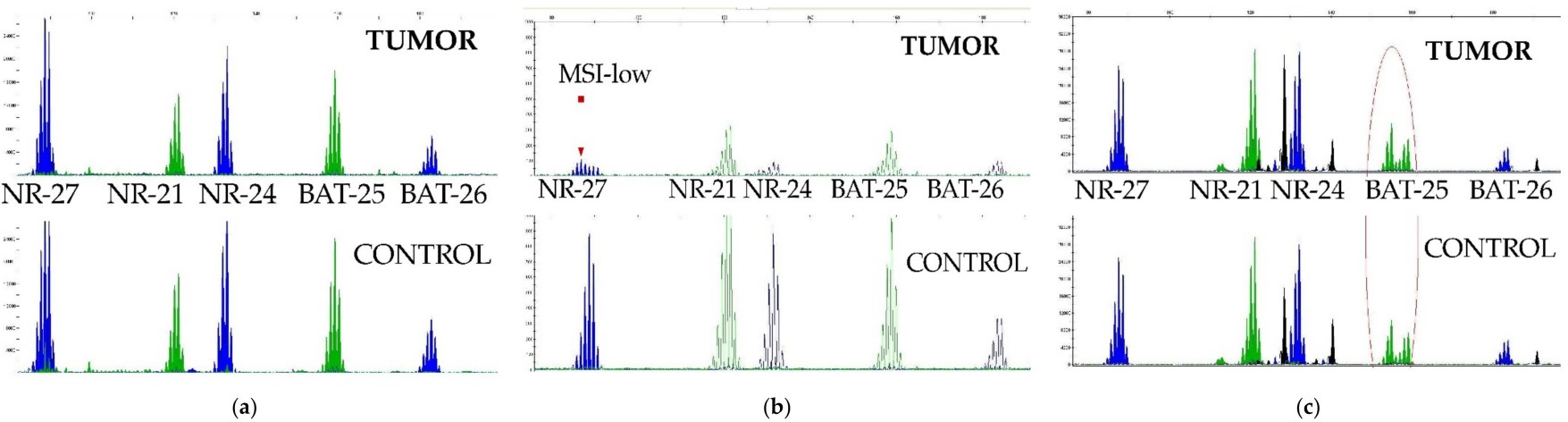

**Figure 4.** Variants of MSI patterns in STR profiles of PMBCL patients: (**a**) MSS, (**b**) MSI-low with instable NR-27 marker, (**c**) MSS with hereditary heterozygous variant of quasi-homozygous BAT-25 marker.

Patients who died from disease progression had neither a stable STR profile nor a LOH-high profile in this cohort (Table 2). It could be speculated that the patients who died, all with an aberrant STR profile, might have been the candidates for ICT salvage therapy if the study was not retrospective. Of the presented cohort, only one patient with an aggressive course of the disease and EMAST microsatellite instability at two STR loci received salvage therapy with the inclusion of nivolumab in the treatment regimen. Achieved remission was maintained for over 20 months.

**Table 2.** LOH and EMAST patterns in the patients undergoing R-Da-EPOCH-21, who died from disease progression (6 from 58 patients, 10.3%).

| Patient | LOH | EMAST | STR Aberrations |
|---------|-----|-------|-----------------|
| #3 | no | 2p | EMAST |
| #6 | 13q | 2p | LOH + EMAST |
| #7 | 5q | 1q, 6q, 7q | LOH + EMAST high |
| #17 | 2p | no | LOH |
| #47 | no | 4q, 13q | EMAST |
| #51 | 12p | 6q | LOH + EMAST |

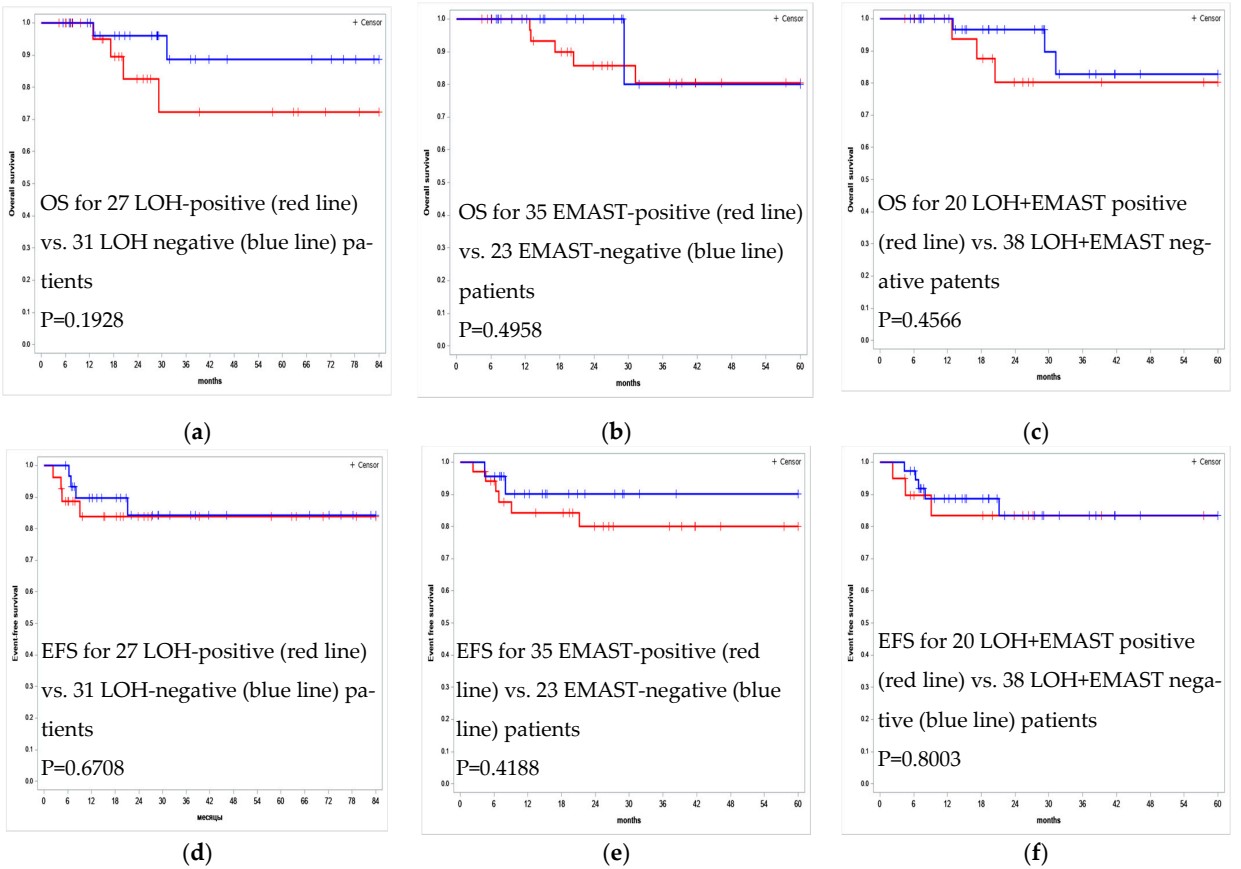

**Figure 5.** Kaplan–Meier survival curves for OS and EFS estimates according to the LOH and EMAST status in tumor STR profile for 58 PMBCL patients undergoing R-Da-EPOCH-21 therapy. (**a**) OS for 27 LOH-positive vs. 31 LOH negative patients (**b**) OS for 35 EMAST-positive vs. 23 EMAST-negative patients (**c**) OS for 20 LOH + EMAST positive vs. 38 LOH + EMAST negative patents (**d**) EFS for 27 LOH-positive vs. 31 LOH-negative patients (**e**) EFS for 35 EMAST-positive vs. 23 EMAST-negative patients (**f**) EFS for 20 LOH + EMAST positive vs. 38 LOH + EMAST negative patients.

The LOH and EMAST status of patients (yes/no test) was not associated with OS and EFS (Figure 5). Unfortunately, our sample size was insufficient to establish a statistically

significant relationship between aberrations in the STR markers and the outcome of standard therapy. With a hypothetical doubling of the sample size and preserving the ratio of stable or aberrant STR profiles, in groups of patients who died or survived, an aberrant STR profile might turn out to be a significant risk factor; however it should take another two or three years to double our sample size since PMBCL is extremely rare (3–4% of non-Hodgkin's lymphomas).

OS at 24 months was 100% in the group with a LOH-high status, compared to 87% in the control group ($p = 0.26$), while EFS was 100% and 80% ($p = 0.16$), respectively. No statistically significant differences in the OS and EFS scores were found between the groups with different LOH-high statuses. Kaplan–Meier survival curves for the EFS and OS estimates according to the LOH-high and EMAST-high status in the tumor STR profile are presented in Figure 6.

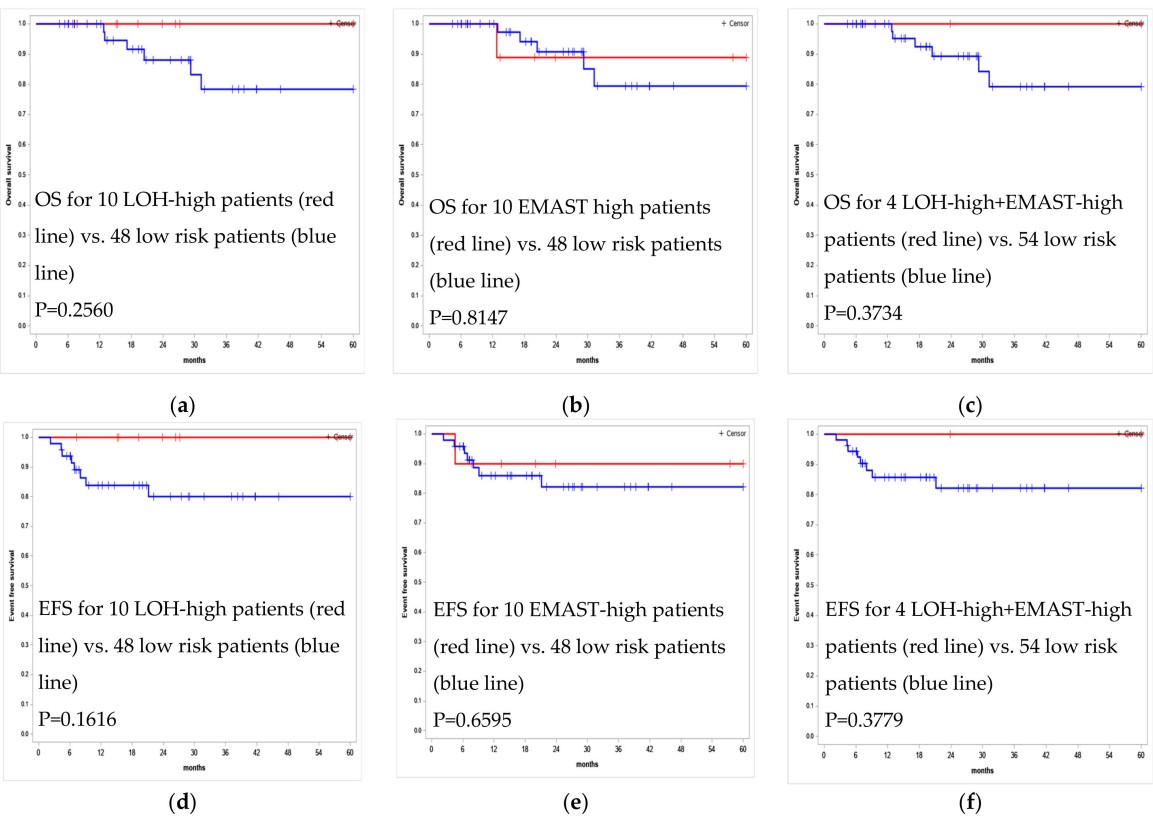

**Figure 6.** Kaplan–Meier survival curves for OS and EFS estimates according to the LOH-high and EMAST-high status in tumor STR profile for 58 PMBCL patients undergoing R-Da-EPOCH-21 therapy. (**a**) OS for 10 LOH-high patients vs. 48 low risk patients (**b**) OS for 10 EMAST high patients vs. 48 low risk patients (**c**) OS for 4 LOH-high + EMAST-high patients vs. 54 low risk patients (**d**) EFS for 10 LOH-high patients vs. 48 low risk patients (**e**) EFS for 10 EMAST-high patients vs. 48 low risk patients (**f**) EFS for 4 LOH-high + EMAST-high patients vs. 54 low risk patients.

## 4. Discussion

The introduction of checkpoint inhibitors has opened up new perspectives in cancer therapy; however, the majority of patients with solid tumors may not benefit from a checkpoint blockade [23–25]. Conversely, the response rate to a PD-1 blockade in relapsed/refractory classic Hodgkin's lymphoma (cHL) ranges from 65% to 84%, which is the highest among all types of malignancies. Currently, checkpoint inhibitors are approved for the treatment of cHL and PMBCL (45% response rate to ICI therapy [12]), but the effectiveness of immunotherapy in other hematological malignancies is extremely low [26].

Due to the uncertain success of immunotherapy, the search for prognostic markers of the ICI response seems to be extremely urgent for PMBCL.

Few studies of microsatellite instability in non-Hodgkin's lymphomas have been reported so far. Gamberi et al. (1997) note the rare occurrence of the MSI phenomenon in NHL; however, PMBCL was not included in this study [27]. Other authors report microsatellite instability for diverse loci in B-cell lymphomas; however, the association of the phenomenon with the outcome of therapy was not tested, and PMBCL was also not studied [28–30].

Sychevskaya et al., in a recent study, described MSI, EMAST, and LOH occurrence in follicular lymphoma and assessed the association of these factors with overall and event-free survival. No statistically significant differences in the OS and EFS scores between groups depending on their MSI, EMAST and LOH status were found [31]. In our study, LOH-high appeared to be a factor of a favorable prognosis in PMBCL; however, this hypothesis requires confirmation from extended patient samples.

EMAST-high is known to be associated with MSI-high in solid tumors [21]. Here we failed to find MSI-high for any of the PMBCL patients. The high frequency of LOH and EMAST in the absence of the MSI phenomenon may be due to an MSH3 deficiency, which is responsible only for the repair of 2–4 nucleotide loops and DNA double-strand breaks (DSB). DNA mismatch repair (MMR) deficiencies lead to MSI, EMAST, and double-strand breaks, while two heterodimeric MutS homolog (MSH) complexes, consisting of either MSH2-MSH6 (MutSα) or MSH2-MSH3 (MutSβ), are responsible for the recognition of these mismatched bases. MSH2-MSH6 binds to single mismatches and small insertions/deletions, whereas MSH2-MSH3 is responsible for ≥2 base loops and DSB repair [32]. Double-stranded DNA breaks lead to the loss of a chromosome fragment and further DNA repair based on a homologous chromosome, therefore, resulting in mosaic copy neutral loss of heterozygosity [33]. Loss of heterozygosity, in turn, may explain the partial loss of tumor antigenicity and escape from the immune response [34]. Meanwhile, further investigation of the mutational status of the MSH3 gene might reveal the nature of such a combination of LOH, EMAST, and MSI in PMBCL.

It could be speculated that in the absence of the MSI-high phenomenon, the response to ICI therapy might reflect the modulation of PD-1 expression on T cells of the tumor microenvironment and PDL-1 on the tumor cells themselves, as well as the high mutational load of the tumor. Thus, three factors—MSI-high, PD-1/PDL-1 expression, and tumor mutational burden—are considered as independent predictors of a response to ICI therapy for solid tumors [35]. STRs were found to be an alternative marker for hypermutability evaluation in different solid tumor types [36] and here we report an initial study towards molecular characterizing of PMBCL as a potential target for immunotherapy. STR profiling of the PMBCL tumor genome revealed LOHs associated with chromosomal gains and losses as well as a copy-neutral loss of heterozygosity. Thus, the LOH frequency reflects the mutational burden level in PMBCL. Detected EMAST seems to be attributed to a failure of the DNA mismatch repair. The high incidence of LOH and EMAST in PMBCL is not associated with standard therapy failure but its role in tumor sensitivity to immunotherapy is still unclear and needs future investigation in combination with the tumor mutational burden and PD-1/PDL-1 level evaluation.

## 5. Conclusions

Given the high survival rate of patients with PMBCL on modern therapy protocols, the main task in routine clinical practice is to de-escalate standard chemotherapy by including targeted drugs. The pathogenesis of PMBCL indicates the possibility of using ICI therapy; however, the molecular predictors of its effectiveness have not been determined yet. Here we report the data of a retrospective study of tumor genome instability markers performed on a limited cohort of PMBCL patients ($n$ = 72). According to our results, microsatellite instability of mononucleotide repeats (MSI-high) practically does not occur in PMBCL. Alternatively, LOH and EMAST should further be considered as putative predictors of a

response to ICI therapy because they are associated with MMR deficiency and occur in half of PMBCL patients. We plan to use our data in a prospective study to include ICI as a first-line therapy in combination with a chemotherapy protocol.

**Author Contributions:** Conceptualization, N.R., Y.M. and A.S.; methodology, N.R.; formal analysis, J.C., and S.K. (Sergey Kulikov); investigation, E.N., Y.K., A.Y. and N.R.; resources, Y.M., A.M., H.J. and S.K. (Sergey Kravchenko); writing—original draft preparation, N.R. and Y.K.; writing—review and editing, A.S.; supervision, A.S.; funding acquisition, N.R. All authors have read and agreed to the published version of the manuscript.

**Funding:** This study was supported by RAKFOND (THE FOUNDATION FOR CANCER RESEARCH SUPPORT, Russia) grant 2.2020.

**Institutional Review Board Statement:** The study was approved by the Ethics Committee of the NATIONAL RESEARCH CENTER for HEMATOLOGY (protocol #157/02.09.2021) and conducted in accordance with the Declaration of Helsinki of 1975, as revised in 2008.

**Informed Consent Statement:** Informed consent was obtained from all subjects involved in the study. Written informed consent has been obtained from the patient(s) to publish this paper.

**Data Availability Statement:** The data presented in this study is available on request from the corresponding author.

**Acknowledgments:** We would like to thank our colleagues from the neighboring laboratory Vadim Surin, Olesya Pshenichnikova, and Yulia Poznyakova for sharing with us the sequencing data concerning MSH3 gene mutation in one of the patients.

**Conflicts of Interest:** The authors declare no conflict of interest.

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
