# Peer review of "STR Profiling Reveals Tumor Genome Instability in Primary Mediastinal B-Cell Lymphoma"

_curroncol, doi:10.3390/curroncol29050278_

Round 1
Reviewer 1 Report
This is a study looking at the genetic lesions in patients with primary mediastinal lymphoma and the possible association with the clinical outcome
The authors looked at the tumor genomes of 72 patients undergoing high-dose chemotherapy treatment. Isolated from the DNA profile for LOH and EMAST by PCR. They found LOH in 37 patients and EMAST in 40, while 24 had both. And this was not associated with worse outcomes.
Adequate introduction and stating of the unmet data are stated
The materials and methods are clear
A clear definition of the high and low occurrences was coupled with very descriptive and informative figures.
The data although is informative, the low number of patients makes it hard to trust the results, when it comes down to a few patients here and there.
Another weakness is the fact that the discussion is very ambitious about the high mutational load of the tumor that could produce independent predictors of the therapy, then they go on to cite that in their patients it was not a predictor of failure to therapy in PMBCL.
Also, the premise about the checkpoint inhibitors and the association with EMSAT and LOH is not looked at in this group but rather the relation with standard treatment.
That leaves us with a disconnect between the aim and what was reported.
Having said that it is a work reporting effort and it is the first block to building the knowledge in regard to genetic instability and its impacts on response to therapy
Reviewer 2 Report
This paper evaluates the incidence of tumor genomic instability in primary mediastinal B-cell lymphoma (PMBCL) with the aim of finding molecular markers that can predict the response to immune checkpoint therapy in patients with PMBCL. I have some questions and suggestions for this article.
- Please double check the insertion of the article references.
- Please explain this abbreviation when it first appears in the article and standardize the article's abbreviation format.
- Please check the article again for spelling and formatting errors.
- Please place the pictures in the right place to make the layout more logical.
- It is recommended to add patient baseline information.
- Table I lists the main characteristics of patients treated with R-Da-EPOCH-21 therapy and LOH and EMAST status, please express the rationale for the selection of these parameters.
- It is recommended to add a description of immune checkpoint treatment options related to this article.
- This paper is a single-center study, whether it is possible to use multi-center data to expand the sample size.
- The conclusion of the article is limited, please briefly describe the innovation of the article.
Author Response
Response to reviewer 2
We would like to acnowledge valuable review that for sure should help us to improve the manuscript. Please find below our point by point comments to the reviewer's questions and suggestions (marked in red)
This paper evaluates the incidence of tumor genomic instability in primary mediastinal B-cell lymphoma (PMBCL) with the aim of finding molecular markers that can predict the response to immune checkpoint therapy in patients with PMBCL. I have some questions and suggestions for this article.1.Please double check the insertion of the article references.
Done.
2.Please explain this abbreviation when it first appears in the article and standardize the article's abbreviation format.
Lacking explanations are added and made uniform.
3.Please check the article again for spelling and formatting errors.
Done.
4.Please place the pictures in the right place to make the layout more logical.
More text descriptions now added just before the references to the figures.
5.It is recommended to add patient baseline information.
Added at lines 94-95
6.Table I lists the main characteristics of patients treated with R-Da-EPOCH-21 therapy and LOH and EMAST status, please express the rationale for the selection of these parameters.
The main idea was to demonstrate the characteristics of patients and evaluate the association of molecular status with selected parameters (gender, age, tumor volume, standard international prognostic index, etc.). These parameters are conventional for characterizing PMLCL patients. Statistical analysis showed that molecular factors are not associated with these clinical parameters.
7.It is recommended to add a description of immune checkpoint treatment options related to this article.
Of the presented cohort, only one patient with an aggressive course of the disease and EMAST microsatellite instability received salvage therapy with the inclusion of nivolumab in the treatment regimen. The achieved remission lasts more than 20 months. Appropriate text is added to the manuscript (lines 166-171).
8.This paper is a single-center study, whether it is possible to use multi-center data to expand the sample size.
In the oncoming project we plan to include other centers into a multicenter study. We have enough facilities for the analysis of a large volume of samples. However coordination of clinical protocols and sample delivery logistics have to be elaborated.
9.The conclusion of the article is limited, please briefly describe the innovation of the article.
A “Conclusion” section is added to the manuscript. (Lanes 249-256)
Round 2
Reviewer 2 Report
- Please double check the insertion of references in the article, for example, in the "Introduction" section, I do not see the insertion of references 1-4, 8 and 9.
- Please check for spelling errors. For example, in line 59, "cells [ MSI-high" and in the title of table1, "theirLOH", please check the contents of the manuscript carefully.
- I recommend aligning the images in the article with the text.
- I recommend explaining the acronym the first time it appears and continuing to use its abbreviated form in subsequent content, rather than using it ambiguously.
